# *Lost in Translation, Found in Spans*:
# Identifying Claims in Multilingual Social Media

**Shubham Mittal**[α†]  **Megha Sundriyal**[α, β†]  **Preslav Nakov**[α]

[α] Mohammed Bin Zayed University of Artificial Intelligence
[β] Indraprastha Institute of Information Technology Delhi
shubhamiitd18@gmail.com, meghas@iiitd.ac.in, preslav.nakov@mbzuai.ac.ae

## Abstract

Claim span identification (CSI) is an important step in fact-checking pipelines, aiming to identify text segments that contain a check-worthy claim or assertion in a social media post. Despite its importance to journalists and human fact-checkers, it remains a severely understudied problem, and the scarce research on this topic so far has only focused on English. Here we aim to bridge this gap by creating a novel dataset, X-CLAIM, consisting of 7K real-world claims collected from numerous social media platforms in five Indian languages and English. We report strong baselines with state-of-the-art encoder-only language models (e.g., XLM-R) and we demonstrate the benefits of training on multiple languages over alternative cross-lingual transfer methods such as zero-shot transfer, or training on translated data, from a high-resource language such as English. We evaluate generative large language models from the GPT series using prompting methods on the X-CLAIM dataset and we find that they underperform the smaller encoder-only language models for low-resource languages.[1]

## 1 Introduction

Social media platforms have become a prominent hub for connecting people worldwide. Along with the myriad benefits of this connectivity, e.g., the ability to share information instantaneously with a large audience, the spread of inaccurate and misleading information has emerged as a major problem (Allcott and Gentzkow, 2017). Misinformation spread via social media has far-reaching consequences, including the potential to sow chaos, to foster hatred, to manipulate public opinion, and to disturb societal stability (Wasserman and Madrid-Morales, 2019; Dewatana and Adillah, 2021).

---

[†]Major part of this work was done during a research internship at MBZUAI.

[1]We release our X-CLAIM dataset and code at https://github.com/mbzuai-nlp/x-claim

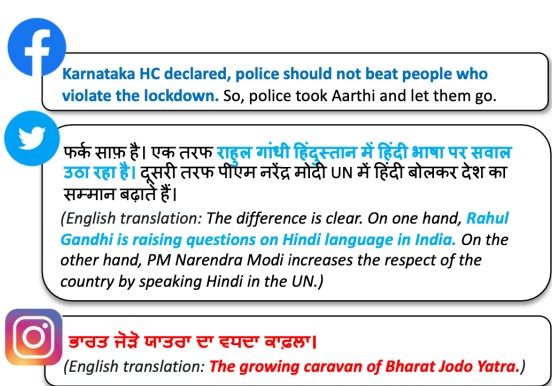

Figure 1: Social media posts from our X-CLAIM dataset. The English translation is shown in parentheses for the Hindi tweet (middle) and for the Punjabi Instagram post (bottom). The claim spans are in bold.

Claims play an integral role in propagating fake news and misinformation, serving as the building blocks upon which these deceptive narratives are formed. In their *Argumentation Theory*, Toulmin (2003) described a claim as "*a statement that asserts something as true or valid, often without providing sufficient evidence for verification.*" Such intentional or unintentional claims quickly gain traction over social media platforms, resulting in rapid dissemination of misinformation as was seen during recent events such as the COVID-19 pandemic (van Der Linden et al., 2020) and Brexit (Bastos and Mercea, 2019). To mitigate the detrimental impact of false claims, numerous fact-checking initiatives, such as PolitiFact and Snopes, dedicate substantial efforts to fact-checking claims made by public figures, organizations, and social media users. However, due to the time-intensive nature of this process, many misleading claims dodge verification and remain unaddressed. To address this, computational linguistic approaches have been developed that can assist human fact-checkers (Vlachos and Riedel, 2014; Nakov et al., 2018; Shaar et al., 2020; Gupta and Srikumar, 2021; Nakov et al., 2021a; Shaar et al., 2022).

Recently, Sundriyal et al. (2022a) introduced the task of claim span identification (CSI), where the goal is to identify textual segments that contain claims or assertions made within the social media posts. The CSI task serves as a precursor to various downstream tasks such as claim verification and check-worthiness estimation.

While efforts have been made in combating misinformation in different languages (Jaradat et al., 2018; Barrón-Cedeño et al., 2023), research in identifying the claim spans has so far been limited to English. Previously, Sundriyal et al. (2022a) have manually extracted COVID-19 claim spans from Twitter in English. However, the landscape of fraudulent claims goes beyond COVID-19 and Twitter. In this work, we aim to bridge these gaps by studying the task of multilingual claim span identification (mCSI) across numerous social media platforms and multiple languages. To the best of our knowledge, this is the first attempt towards identifying the claim spans in a language different from English.

We design the first data curation pipeline for the task of mCSI, which, unlike Sundriyal et al. (2022a), does not require manual annotation to create the training data. We collect data from various fact-checking sites and we automatically annotate the claim spans within the post. Using the pipeline, we create a novel dataset, named X-CLAIM, containing 7K real-world claims from numerous social media platforms in six languages: English, Hindi, Punjabi, Tamil, Telugu, and Bengali. Figure 1 showcases a few examples from our dataset.

We report strong baselines for the mCSI task with state-of-the-art multilingual models. We find that joint training across languages improves the model performance when compared to alternative cross-lingual transfer methods like zero-shot transfer, or training on translated data, from a high-resource language like English. In this work, we make the following contributions:

- We introduce the first automated data annotation and curation pipeline for the mCSI task.

- We create a novel dataset, named X-CLAIM, for the mCSI task in six languages.

- We experiment with multiple state-of-the-art encoder-only language models and the generative large language models to achieve high performance on the proposed task.

## 2 Related Work

Efforts to combat misinformation and fake news have focused on claims in various sources. The existing body of work in this area can be broadly categorized into the following major groups: claim detection (Chakrabarty et al., 2019; Gupta et al., 2021; Wührl and Klinger, 2021; Gangi Reddy et al., 2022a,b), claim check-worthiness (Jaradat et al., 2018; Wright and Augenstein, 2020), claim span identification (Sundriyal et al., 2022a), and claim verification (Ma et al., 2019; Soleimani et al., 2020). Being the precursor of several other downstream tasks, claim detection has garnered significant attention. Various methods have been proposed to tackle claim detection, aiming to identify statements that may contain claims (Lippi and Torroni, 2015; Levy et al., 2017; Gangi Reddy et al., 2022b). In response to the escalating issue of false claims on social media, there has been a surge in the development of claim detection systems specifically designed to handle text from social media platforms (Chakrabarty et al., 2019; Gupta et al., 2021; Sundriyal et al., 2021). Recently, Sundriyal et al. (2022a) introduced the task of claim span identification where the system needs to label the claim-containing textual segments from social media posts, making claim detection systems more explainable through this task.

While most existing methods to combat fake news are primarily tailored for English (Levy et al., 2014; Lippi and Torroni, 2015; Sundriyal et al., 2021, 2022b), in recent times, there has been a surge in interest regarding the advancement of fact-checking techniques for various other languages. ClaimRank (Jaradat et al., 2018) introduced an online system to identify sentences containing check-worthy claims in Arabic and English. The Check-That! Lab has organized several multilingual claim tasks over the past five years, progressively expanding language support and garnering an increasing number of submissions (Nakov et al., 2018; Elsayed et al., 2019; Shaar et al., 2020; Nakov et al., 2021b, 2022). In their latest edition, Barrón-Cedeño et al. (2023) featured factuality tasks in seven languages: English, German, Arabic, Italian, Spanish, Dutch, and Turkish. Gupta and Srikumar (2021) introduced X-FACT, a comprehensive multilingual dataset for factual verification of real-world claims in 25 languages. Unlike that work, here we focus on extracting the claim from a social media post, rather than fact-checking a claim.

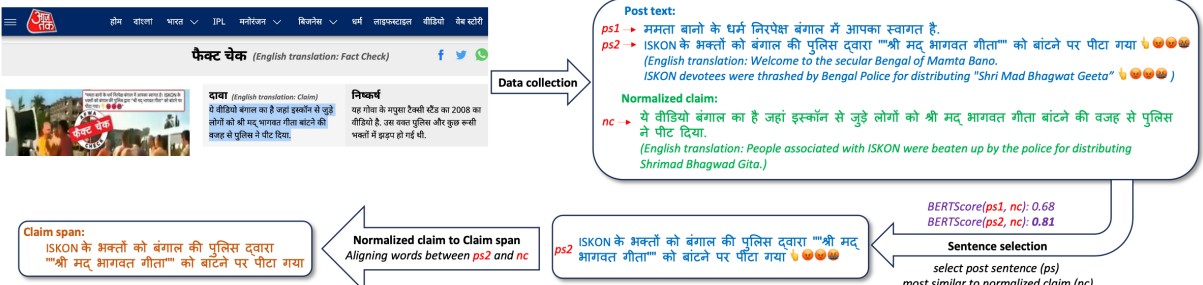

Figure 2: Our two-step methodology to create the X-CLAIM dataset for the multilingual claim span identification task. The top row shows the data collection (Section 3.1) from a fact-checking website. The bottom row illustrates the automated annotation step (Section 3.2) in which, first, the most similar post sentence (*ps*) is selected, and then, the claim span is created with the help of a normalized claim (*nc*). We use BERTScore-Recall (Zhang et al., 2020) for sentence selection and awesome-align (Dou and Neubig, 2021) for word alignment between *nc* and *ps2*.

The task of claim span identification remains unexplored due to the lack of datasets in other languages. Sundriyal et al. (2022a) developed a dataset of 7.5K manually annotated claim spans in tweets, named CURT; all the tweets and claim spans in that dataset are in English. Additionally, while there has been interest in claims in other languages, there is a notable lack of progress on Indian languages. Here, we aim to bridge this gap.

## 3 Dataset

We follow a two-step pipeline to develop our dataset: (*i*) data collection and (*ii*) automated annotation. We present a high-level overview of our proposed data creation methodology in Figure 2. Below, we explain these steps in detail.

### 3.1 Data Collection

We observe in various fact-checking websites that professional fact-checkers, while investigating a given social media post or news article, first find the claim made in the post, which we call a *normalized* claim, and then they verify whether that claim is true, misleading, or false. This is the motivation for the CSI task as a precursor to fact-checking as it is a step in the fact-checking process as performed by humans. Thus, we leverage the efforts of fact-checkers and we collect data from numerous fact-checking websites that are recognized by the International Fact-Checking Network (IFCN).[2] We aim to create a dataset comprising claims made in social media and in multiple languages, with a focus on Indian languages. We scrape data from fact-checked posts in six languages: English, Hindi, Punjabi, Tamil, Telugu, and Bengali.

We highlight that we deal with low-resource languages since we found only a couple of fact-checking websites that analyze social media posts in languages other than English. For each website, we scrape all the fact-checked posts[3] with the help of a web scraping API.[4]

Then, we collect the text of the social media post text and the normalized claim from the web page of each fact-checked post with the help of regular expressions based on the structure of the fact-checking website. Finally, we use various filtering rules to remove posts that are about videos, Instagram reels, or when their text is too short or excessively long. These rules help us to collect only the social media posts with a text modality. We provide more details about the process of data collection in Appendix A.

### 3.2 Automated Annotation

We label the claim-containing a textual segment within the social media post using the human-written normalized claim as a guidance from the previous step. The normalized claim can be relied on to be extremely trustworthy since it was manually written by professional fact-checkers. However, it does not have to be literally spelled out as part of the social media post. Having this normalized claim gives us a good guidance about where to look for the claim span, and we try to do this mapping automatically.

As shown in the bottom row in Figure 2, this step includes two substeps: sentence selection and conversion of the normalized claim to the claim span. Both substeps use modules that support multiple languages and do not require human intervention.

---

[2]https://www.poynter.org/ifcn/

[3]The data was scraped in May 2023.

[4]https://www.octoparse.com/

|              | English (En)       | Hindi (Hi)          | Punjabi (Pa)        | Tamil (Ta)          | Telugu (Te)         | Bengali (Bn)        |
|--------------|--------------------|---------------------|---------------------|---------------------|---------------------|---------------------|
| # train      | 3891               | 1193                | 346                 | 100                 | -                   | -                   |
| # dev        | 400                | 100                 | 100                 | 30                  | -                   | -                   |
| # test       | 371                | 100                 | 100                 | 100                 | 107                 | 102                 |
| text len (t) | $37.58_{\pm34.59}$ | $28.59_{\pm23.07}$  | $29.00_{\pm21.92}$  | $26.40_{\pm20.10}$  | $24.42_{\pm15.17}$  | $29.48_{\pm21.73}$  |
| claim len (t)| $17.67_{\pm12.33}$ | $17.79_{\pm11.62}$  | $17.10_{\pm11.20}$  | $14.12_{\pm08.27}$  | $13.54_{\pm06.73}$  | $15.00_{\pm07.69}$  |
| text len (c) | $229.34_{\pm208.56}$ | $143.05_{\pm114.19}$ | $145.03_{\pm106.75}$ | $229.63_{\pm174.83}$ | $186.63_{\pm113.99}$ | $186.95_{\pm132.81}$ |
| claim len (c)| $108.95_{\pm81.88}$ | $85.25_{\pm56.54}$  | $85.25_{\pm53.71}$  | $122.23_{\pm68.86}$ | $104.50_{\pm50.65}$ | $97.42_{\pm46.71}$  |

Table 1: Statistics about the X-CLAIM dataset. The number of samples in the train, the development, and the test splits are reported in first three rows, respectively. Text len and claim len are the average ($\pm$ standard deviation) length of social media post text and claim span, respectively, in number of tokens ($t$) and characters ($c$), respectively.

First, we look for the most relevant sentence that encapsulates the claim made in the post. We do this by computing a similarity score between the normalized claim and each of the post's sentences, and we select the sentence with the maximum score.

Second, using awesome-align (Dou and Neubig, 2021), we find the word tokens in the post sentence that align with the word tokens in the normalized claim. We then obtain the claim span as the sequence of word tokens, starting with the first aligned word token and ending with the last aligned word token in the sentence.

We use Stanza (Qi et al., 2020) to perform sentence segmentation for English, Hindi, Tamil, and Telugu. For Punjabi and Bengali, we consider the complete post text as a single sentence since we did not find any publicly available sentence segmentation tools for these languages. While using awesome-align in conversion from the normalized claim to the claim span, we used the official repository of Dou and Neubig (2021). Recent works (Yarmohammadi et al., 2021; Kolluru et al., 2022) have used word-alignment to produce silver labels in the target language (like Hindi) using gold labels available in the source language (like English). Mittal et al. (2023) used word alignments from awesome-align, and then considered the longest contiguous sequence of aligned tokens in the translated text as the final projected gold labels. Taking the longest contiguous sequence is suitable for tasks where the target text, the gold labels, or both, are relatively short. However, in our mCSI task, the normalized claims and the post texts are quite long (see Table 1). Thus, we took the sequence of words from the first to the last aligned word. We found that this yielded better performance than taking the longest contiguous sequence of aligned words in the social media post.

Note that we empirically chose the most appropriate sentence similarity measure for sentence selection, after trying a variety of similarity measures. Tasks such as machine translation (Dong et al., 2015) and text summarization (Liu and Lapata, 2019) require evaluation measures that take paraphrasing and synonyms into account while comparing the model's generated text to the gold reference text. We leverage these evaluation measures for sentence similarity. To evaluate the commonly used measures such as ROUGE (Lin, 2004), METEOR (Banerjee and Lavie, 2005) and BERTScore, we manually annotated the claim spans for 300 randomly sampled posts in the six languages. Then, we evaluated the automatically annotated claim spans when using different similarity measures against the manually annotated claim spans. The results are shown in Table 2: we can see that BERTScore-Recall yields consistently better performance for finding the annotated spans. For Punjabi and Bengali, we only used awesome-align due to the lack of a sentence segmentation module and we observed high-quality F1 scores of 81.23% and 78.6%, respectively.

Overall, our two-step data creation methodology yields a robust, scalable, and high-quality automatically annotated data for our multilingual claim span identification task.

| Approach         | En        | Hi        | Ta        | Te        |
|------------------|-----------|-----------|-----------|-----------|
| awesome-align    | 70.48     | 77.27     | 82.24     | 82.61     |
| +ROUGE-F1        | 74.19     | **80.52** | 82.62     | 78.85     |
| +METEOR          | 78.71     | 79.50     | **82.64** | 81.08     |
| +BERTScore-F1    | 79.60     | **80.52** | **82.64** | 82.73     |
| +BERTScore-Recall| **83.91** | **80.52** | **82.64** | **83.34** |

Table 2: F1 score (in %) of the automated annotation for our data curation pipeline when using different sentence similarity measures during sentence selection.

## 3.3 Evaluation Sets and Dataset Analysis

We created the evaluation sets with the help of linguistic experts in the six languages. We provided them with nearly 100 samples from the curated data in each language (400 in English) along with detailed annotation guidelines for the CSI task from Sundriyal et al. (2022a). We asked them to annotate the claim spans in the social media posts under the guidance of claims authored by professional fact-checkers. We created training and development splits in a ratio of 80:20 on the remaining curated data. For Telugu and Bengali, we only formed test sets as there were less examples available for these languages. Table 1 shows statistics about the dataset and the splits, and Figure 1 shows a few examples from our X-CLAIM dataset.

Table 1 further reports the length of the post text and the claim span. As the claim spans are generally concise and do not contain extra neighboring words, we observe that the claim spans are nearly half of the text of the post for all languages.

## 4 Experiments

**Evaluation Measures:** Following Sundriyal et al. (2022a), we address mCSI as a sequence tagging task. For evaluation, we use three measures, computed at the span level (Da San Martino et al., 2019): Precision (P), Recall (R), and F1-score.

**Models:** We use state-of-the-art transformer-based (Vaswani et al., 2017) multilingual pre-trained encoder-only language models such as mBERT (Devlin et al., 2019), mDeBERTa (He et al., 2023), and XLM-RoBERTa (XLM-R) (Conneau et al., 2020). We encode each post's token with *IO (Inside-Outside)* tags to mark the claim spans. Other encodings such as *BIO*, *BEO* and *BEIO* performed worse (see Appendix C for detailed comparison of encodings). More details about the training are given in Appendix B.

## 5 Results

We carry out an exhaustive empirical investigation to answer the following research questions:

R1. Does the model benefit from joint training with multiple languages? (Section 5.1)

R2. Do we need training data in low-resource languages when we have abundant data in high-resource languages?[5] (Section 5.2)

R3. Can large language models (LLMs) such as GPT-4 identify the claims made in multilingual social media? (Section 5.3)

R4. How does the automatically annotated X-CLAIM dataset compare to prior manually annotated datasets like CURT? (Section 5.4)

## 5.1 Training on Multilingual Social Media

We train and compare two kinds of models: Monolingual and Multilingual models. In a Monolingual setup, we train one model for each language using the available training data in X-CLAIM dataset, whereas in a Multilingual setup, we train a single model on the training data for all languages combined. We note that there is no Monolingual model for Telugu and Bengali due to the lack of training data for these languages. However, we evaluate the Multilingual model on them as that model was trained in multiple languages. The performance of these models with different pretrained encoders is shown in Table 3.

We can see that the Multilingual models outperform the Monolingual models by 1.15% precision and 0.93% F1, averaged over all languages (except for Telugu and Bengali). Even though the recall gets hurt by 0.45%, the improvement in F1 suggests that the model does benefit from joint training. We posit that the drop in recall and the gain in precision indicate that the model has become more careful when identifying the claims.

## 5.2 Cross-lingual Transfer from English

We use the English training data in two experimental settings and we compare them to Multilingual models. In the first setting, we leverage the strong cross-lingual transfer capabilities of pretrained multilingual models (Wu and Dredze, 2019). We take Monolingual models for English and test them on the remaining five languages. In this setting, we have zero-shot transfer from monolingual-English models. In the second setting, which we call *translate-train* models, we translate the English training data to the target language and we train a model only on the translated data. To perform translation of social media posts, we use Google translate,[6] and we project the claim spans (in English), or the token labels, on the translated post using our automated annotation pipeline (see Section 3.2 for detail).

---

[5]We consider English to be a high-resource language.

[6]https://translate.google.com/

| Model | English | | | Hindi | | | Punjabi | | | Tamil | | | Telugu | | | Bengali | | |
|---|---|---|---|---|---|---|---|---|---|---|---|---|---|---|---|---|---|---|
| | P | R | F1 | P | R | F1 | P | R | F1 | P | R | F1 | P | R | F1 | P | R | F1 |
| *monolingual models (train using only training data in target language)†* | | | | | | | | | | | | | | | | | | |
| mBERT | 70.30 | 77.08 | 69.86 | 77.57 | 88.93 | 79.03 | 69.40 | 94.22 | 76.93 | 73.83 | 87.74 | 76.94 | n/a | n/a | n/a | n/a | n/a | n/a |
| mDeBERTa | **74.28** | 80.72 | **73.79** | 75.94 | 92.30 | 79.84 | 69.55 | 92.14 | 75.78 | 67.68 | 76.29 | 69.05 | n/a | n/a | n/a | n/a | n/a | n/a |
| XLM-R | 71.56 | **81.51** | 72.79 | 75.34 | **94.49** | **81.09** | 68.85 | 93.58 | 75.62 | 72.68 | 80.42 | 71.82 | n/a | n/a | n/a | n/a | n/a | n/a |
| *multilingual models (train using training data in all languages)* | | | | | | | | | | | | | | | | | | |
| mBERT | 70.86 | 77.01 | 70.39 | 76.16 | 90.40 | 80.04 | 68.30 | 88.19 | 73.96 | 73.80 | 85.94 | 76.57 | 79.58 | 80.74 | 78.07 | 76.79 | 86.22 | **79.39** |
| mDeBERTa | 72.25 | 80.90 | 73.01 | 75.94 | 92.66 | 80.87 | 70.62 | 90.99 | 76.27 | **78.21** | 88.69 | **80.29** | 82.34 | **87.10** | 82.92 | **77.11** | 85.89 | 79.24 |
| XLM-R | 72.45 | 78.61 | 71.93 | 75.30 | 89.09 | 78.63 | **73.65** | 87.95 | **77.03** | 73.23 | 83.63 | 74.76 | 80.48 | 85.29 | 80.68 | 76.99 | 81.24 | 77.22 |
| *zero-shot transfer from monolingual-English models ‡* | | | | | | | | | | | | | | | | | | |
| mBERT | n/a | n/a | n/a | 74.53 | 83.46 | 76.51 | 66.84 | 79.11 | 69.91 | 76.92 | 72.77 | 70.85 | 79.58 | 68.68 | 70.31 | 72.74 | 80.91 | 74.21 |
| mDeBERTa | n/a | n/a | n/a | 75.71 | 91.18 | 80.08 | 73.18 | 88.87 | 76.78 | 77.68 | 80.97 | 75.44 | **84.42** | 74.91 | 76.25 | 76.42 | 79.49 | 75.88 |
| XLM-R | n/a | n/a | n/a | 73.42 | 88.28 | 77.42 | 70.88 | 92.44 | 76.68 | 76.88 | 78.89 | 74.33 | 80.44 | 79.28 | 77.81 | 73.67 | 80.37 | 75.04 |
| *translate-train models (train on translated training data from English to target language)‡* | | | | | | | | | | | | | | | | | | |
| mBERT | n/a | n/a | n/a | **78.60** | 86.52 | 79.55 | 67.33 | 92.80 | 74.70 | 75.93 | 81.82 | 76.04 | 75.56 | 72.07 | 71.76 | 70.16 | 83.97 | 74.36 |
| mDeBERTa | n/a | n/a | n/a | 76.73 | 87.43 | 78.77 | 68.84 | 91.73 | 75.36 | 77.46 | **89.06** | 80.13 | 82.18 | 73.63 | 75.16 | 72.41 | **88.78** | 77.57 |
| XLM-R | n/a | n/a | n/a | 75.55 | 83.37 | 76.11 | 68.97 | **94.43** | 76.53 | 77.97 | 82.79 | 77.59 | 77.07 | 72.11 | 72.40 | 69.99 | 86.82 | 75.47 |

Table 3: Precision (P), recall (R) and F1 performance (in %) of pretrained encoder-only language models in different settings on X-CLAIM dataset. †The monolingual models for Telugu and Bengali are not available (n/a) due to the lack of training data for these languages. ‡n/a: we do not evaluate on the English test set since we focus on the cross-lingual transfer from English to the target language. The reported numbers are the median of three runs with different seeds as high variance was observed across the fine-tuning runs. The best scores are in bold.

Both the zero-shot transfer and the translate-train models are almost consistently worse than the MULTILINGUAL models (in terms of F1) for all five languages. The translate-train models show a drop of 1.19% F1, whereas zero-shot transfer models are 2.13% F1 behind MULTILINGUAL. This offers strong evidence that the training data in low-resource languages helps over the training data in a high-resource language.

Interestingly, we notice that zero-shot transfer models are consistently worse than translate-train ones when using mBERT and mDeBERTa, for all five languages. For instance, with mBERT, zero-shot transfer models are worse by 2.92% F1. However, with XLM-R, zero-shot transfer models are better than translate-train models by 1.15% precision and 0.64% F1. We believe that this is because XLM-R has stronger cross-lingual transfer capabilities, stemming from its larger pretraining data compared to mBERT and mDeBERTa.

### 5.3 Evaluating the GPT Series LLMs

We experiment with several large language models (LLMs):[7] text-davinci-003 (T-DV3), gpt-3.5-turbo (GPT-3.5) and gpt-4-0314 (GPT-4) on the mCSI task using the OpenAI API.[8] We prompted each LLM with each social post from the test sets in our X-CLAIM dataset and we asked the LLM to respond with the claim span.

[7]https://platform.openai.com/docs/models
[8]https://platform.openai.com/docs/api-reference

The generated response may contain words that are either not present in the post or are synonyms of words from the posts. Thus, we treated the response like a normalized claim (Section 3.2) and we passed it through our automated annotation step (Section 3.2) to create the corresponding claim span. We evaluated the predicted claim spans with respect to the gold claim spans. More details about this setup are given in Appendix D.

**Zero-shot Prompting.** We experiment with four prompts that use no examples: IDENTIFY, EXTRACT, SPAN, and LANGUAGE. The exact prompt structure is given in Figure 5 in the Appendix. Table 4 shows their performance when used with different LLMs on our X-CLAIM dataset.

We noticed that the LLMs mostly responded in English even when asked to analyze a post in another language. One reason could be that the prompts do not explicitly specify the language the LLM should respond in. Since our automated annotation step is language-agnostic, the corresponding claim span is in the target language. To overcome this, we asked the LLM to respond in the target language with the LANGUAGE prompt. Interestingly, and unlike GPT-3.5 and GPT-4, the performance of T-DV3 with LANGUAGE prompt significantly dropped by 12-37% F1 (averaged over all languages except English) when compared to the other three prompts. This suggests that T-DV3 is weaker in a multilingual setup.

| Model | English | | | Hindi | | | Punjabi | | | Tamil | | | Telugu | | | Bengali | | |
|---|---|---|---|---|---|---|---|---|---|---|---|---|---|---|---|---|---|---|
| | P | R | F1 | P | R | F1 | P | R | F1 | P | R | F1 | P | R | F1 | P | R | F1 |
| *IDENTIFY prompt: Identify the central claim* | | | | | | | | | | | | | | | | | | |
| T-DV3 | 70.07 | 60.64 | 61.83 | 70.96 | 63.95 | 60.55 | 67.99 | 92.85 | 72.67 | 71.93 | 48.15 | 48.90 | 73.66 | 42.28 | 46.88 | 66.43 | 84.49 | 67.37 |
| GPT-3.5 | 69.76 | 74.68 | 69.28 | 73.82 | 83.72 | 75.09 | 62.98 | **97.28** | **72.72** | 72.06 | 76.53 | 71.08 | 79.90 | 69.53 | 72.09 | 64.47 | **98.26** | **74.00** |
| GPT-4 | 74.14 | 75.49 | 71.89 | 76.72 | 78.80 | 74.64 | 64.39 | 93.17 | 72.32 | 74.42 | 74.52 | 70.78 | 79.69 | 68.49 | 71.96 | 66.95 | 93.95 | 73.54 |
| *EXTRACT prompt: Extract the central claim* | | | | | | | | | | | | | | | | | | |
| T-DV3 | 70.69 | 61.70 | 63.01 | 77.97 | 36.96 | 41.62 | 65.07 | 75.29 | 58.79 | 76.47 | 35.70 | 38.28 | 74.75 | 31.91 | 36.57 | 72.04 | 47.82 | 43.93 |
| GPT-3.5 | 69.56 | 74.02 | 68.75 | 74.70 | **85.84** | 76.59 | 63.14 | 97.24 | 72.54 | 73.37 | **78.44** | 72.59 | 80.72 | 69.85 | 72.63 | 64.34 | 97.88 | 73.59 |
| GPT-4 | 74.53 | 75.05 | 71.96 | 76.92 | 78.62 | 74.70 | 64.83 | 92.32 | 71.79 | 73.04 | 74.07 | 70.31 | 82.21 | 71.93 | 74.63 | 68.23 | 89.95 | 72.97 |
| *SPAN prompt: Extract the central claim span* | | | | | | | | | | | | | | | | | | |
| T-DV3 | 67.35 | 54.78 | 56.99 | 73.33 | 28.52 | 33.78 | 67.46 | 42.67 | 41.15 | 72.76 | 25.41 | 29.57 | 70.92 | 26.71 | 32.90 | 62.45 | 33.97 | 33.40 |
| GPT-3.5 | 69.05 | 71.48 | 67.07 | 75.44 | 70.91 | 69.23 | 66.81 | 82.25 | 69.03 | 68.53 | 70.53 | 66.39 | 79.35 | 66.60 | 68.96 | 71.47 | 73.40 | 67.78 |
| GPT-4 | **80.79** | 74.19 | **74.46** | **84.99** | 69.32 | 72.92 | **77.39** | 69.87 | 68.45 | **82.58** | 62.71 | 68.24 | **85.56** | 64.49 | 70.84 | **77.18** | 69.07 | 68.60 |
| *LANGUAGE prompt: Extract the central claim in <Language>* | | | | | | | | | | | | | | | | | | |
| T-DV3 | 70.60 | 62.37 | 63.35 | 80.90 | 20.59 | 28.14 | 73.63 | 19.97 | 26.03 | 77.28 | 09.65 | 15.46 | 81.31 | 08.35 | 14.73 | 72.93 | 19.06 | 23.44 |
| GPT-3.5 | 70.70 | **76.65** | 70.49 | 73.61 | 79.61 | 72.97 | 64.04 | 83.52 | 67.77 | 77.81 | 71.24 | 70.46 | 82.44 | 67.72 | 71.43 | 68.90 | 79.64 | 69.94 |
| GPT-4 | 74.76 | 75.66 | 72.41 | 79.51 | 80.87 | **77.27** | 64.21 | 92.31 | 71.23 | 81.47 | 75.38 | **75.32** | 84.96 | **72.75** | **76.15** | 68.95 | 83.50 | 71.14 |

Table 4: Performance (in %) of the LLMs: text-davinci-003 (T-DV3), gpt-3.5-turbo (GPT-3.5) and gpt-4-0314 (GPT-4) on the X-CLAIM dataset using zero-shot prompting. GPT-4 nearly always shows the best performance whereas GPT-3.5 shows consistent and significant gains as compared to T-DV3. Huge performance drops are observed in T-DV3 with LANGUAGE prompt when compared to the remaining three *language-loose* prompts.

We further find that GPT-4 is nearly always better than GPT-3.5 by an average of 4.23% precision and 1.5% F1 over the four prompts. GPT-3.5 consistently outperformed T-DV3 by an average of 35.96% recall and 27.63% F1, but it lags behind by 0.5% in terms of precision.

**In-Context Learning.** Here, we give the model a few labeled examples as part of the prompt as shown in Figure 6 of the Appendix. Since GPT-4 outperformed the other two LLMs and showed the best performance with LANGUAGE (Table 4), we experimented with in-context learning with GPT-4 and LANGUAGE prompt. For Telugu and Bengali, we use examples from translated data (Section 5.2) due to the lack of training data in these languages. The results are shown in Table 5.

We see that in-context learning consistently improves F1 score over the zero-shot prompting in all six languages. With more examples shown, the performance increased in English, Hindi and Punjabi at the cost of more computation time. We find that 10-shot in-context learning improved the performance by an average of 2.78% F1 for the six languages in comparison to zero-shot prompting.

**Comparing mDeBERTa and GPT-4.** We compared the best-performing fine-tuned encoder-only language model to the best-performing generative LLM. The MULTILINGUAL mDeBERTa model and GPT-4 yielded the best results for most languages as reported in Table 3, Table 4, and Table 5.

In the case of GPT-4, the best setting uses the LANGUAGE prompt with 10-shot in-context learning for the six languages. Figure 3 compares the two models in terms of F1 scores; we further offer comparison in terms of precision and recall in Table 11 of the Appendix.

We can see in Figure 3 that MULTILINGUAL mDeBERTa outperforms GPT-4 by 2.07% F1, averaged over the six languages. GPT-4 shows competitive performance with mDeBERTa in English, Hindi and Punjabi. On the remaining three languages, mDeBERTa outperforms GPT-4 by a large margin of 2-7% F1. This suggests that LLMs show strong performance on high-resource languages like English, but still lag behind smaller fine-tuned LMs on low-resource languages such as Bengali.

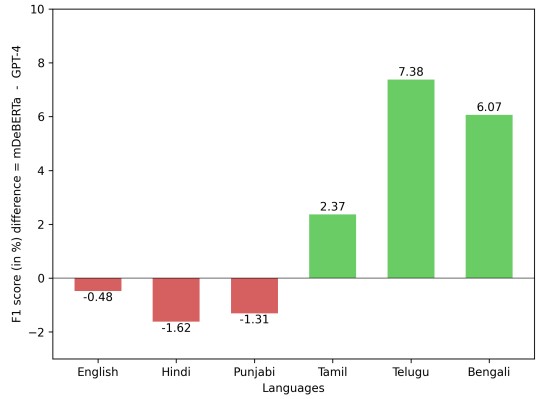

Figure 3: Performance comparison (in %) of the mDeBERTa model in MULTILINGUAL setup and the GPT-4 model on our X-CLAIM dataset.

| # Examples | English | | | Hindi | | | Punjabi | | | Tamil | | | Telugu | | | Bengali | | |
|---|---|---|---|---|---|---|---|---|---|---|---|---|---|---|---|---|---|---|
| | P | R | F1 | P | R | F1 | P | R | F1 | P | R | F1 | P | R | F1 | P | R | F1 |
| 0 | 74.76 | 75.66 | 72.41 | **79.51** | 80.87 | 77.27 | 64.21 | **92.31** | 71.23 | 81.47 | 75.38 | 75.32 | 84.96 | 72.75 | 76.15 | 68.95 | 83.50 | 71.14 |
| 1 | **75.55** | 76.45 | 73.23 | 77.41 | 79.43 | 76.22 | 67.50 | 87.95 | 72.15 | **81.52** | 82.59 | 79.21 | 85.22 | **74.38** | 77.49 | 70.87 | 84.00 | 73.18 |
| 4 | 74.76 | 76.49 | 72.74 | 79.09 | 85.19 | 79.09 | 73.90 | 88.64 | 76.32 | 81.30 | 81.25 | 78.25 | 85.08 | 74.28 | 77.18 | 70.29 | **85.72** | 73.37 |
| 7 | 74.28 | 76.31 | 72.29 | 78.05 | 86.40 | 79.59 | 71.11 | 90.30 | 76.06 | 78.44 | 80.61 | 76.76 | 85.01 | 73.06 | 76.46 | 70.42 | 80.54 | 71.18 |
| 10 | 75.28 | **77.80** | **73.49** | 79.42 | **91.06** | **82.49** | 73.49 | 91.20 | **77.58** | 79.12 | 81.99 | 77.92 | **86.11** | 71.31 | 75.54 | **71.88** | 81.79 | 73.17 |

Table 5: Performance (in %) of the GPT-4 model using in-context learning with LANGUAGE prompt. The first row contains zero-shot prompting (i.e., no examples) results from Table 4. The best scores are in bold.

## 5.4 Comparing X-CLAIM and CURT

We trained mDeBERTa on the CURT dataset (Sundriyal et al., 2022a), containing tweets in English, and we compared it to the English MONOLINGUAL model (trained with mDeBERTa on English data in X-CLAIM) on the test sets for the six languages in the X-CLAIM dataset. We show the F1 scores for both models in Figure 4 and we report the precision and the recall scores in Table 12 in the Appendix.

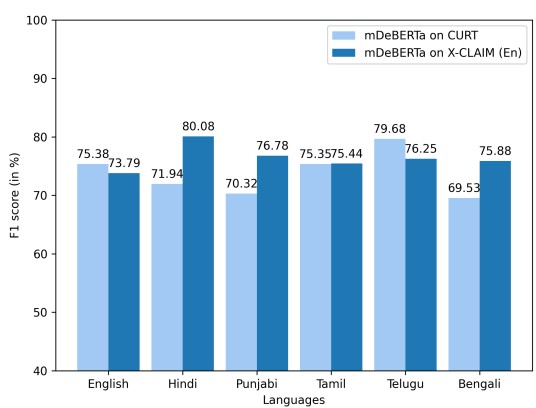

Figure 4: F1 score (in %) of mDeBERTa trained on the CURT dataset (Sundriyal et al., 2022a) vs. mDeBERTa trained on the English data in X-CLAIM dataset.

The mDeBERTa model fine-tuned on the X-CLAIM English data performs competitively in English with the CURT trained model and shows 3.52% F1 average gain over the remaining five languages. Note that CURT is manually annotated and is twice larger than the English part of the X-CLAIM dataset. This offers empirical evidence of better model generalization when training on the X-CLAIM dataset compared to the CURT dataset.

## 6 Error Analysis

In this section, we qualitatively analyze the errors made by the best-performing MULTILINGUAL mDeBERTa model. To provide insights on how LLMs can be improved for this task, we also discuss the errors made by GPT-4 in its best-performing setting of 10-shot in-context learning.

We analyzed the predictions on the test examples in English and Hindi, and we report the kinds of errors made by the two language models in Table 6. Below, we discuss the results of the analysis.

**English.** In the first post in Table 6, both models deviate from the gold claim span. GPT-4 model correctly identifies the presence of the claim but inadvertently veers away from the central check-worthy assertion and focuses on the secondary claim. On the other hand, the mDeBERTa model includes information about *moisture and bacteria in the mask*, but contains several grammatical errors and lacks clarity. In particular, the phrase *'every day day legionnaires disease'* is confusing and doesn't convey a clear message.

Both models provide similar claim spans for the second social media post, capturing the central assertion accurately. However, mDeBERTa contains the extra words *'pregnancy your'* at the beginning that are not present in the gold span. These extra words introduce confusion and do not accurately represent the claim made in the social media post.

**Hindi.** Claim span identification in other languages is more complicated than in English due to the lack of proper guidelines pertaining to their linguistic characteristics. In the first example, GPT-4 almost accurately predicted the span, missing the word 'श्रीमती' in the beginning. While mDeBERTa predicted both the claim and the premise, defying the very purpose of the task, which is to extract precise claim phrases from the post.

In the second post, both models performed well overall. However, we observe a similar issue as for English: the inclusion of additional phrases alongside the claim spans, which can potentially detract from the clarity and precision of the claim. This indicates that these models struggle to make precise decisions about claim boundaries.

We can conclude that for both languages, the models can identify the claim but might propose wider boundaries, including extra words.

| | Test Instances | GPT-4 | mDeBERTa |
|---|---|---|---|
| English | **Post:** Mask wearers beware. A caller to a radio talk show recently shared that his wife was hospitalized and told she had COVID and only a couple of days left to live. A doctor friend suggested she be tested for legionnaires disease because she wore the same mask every day all day long. Turns out it WAS legionnaires disease from the moisture and bacteria in her mask. She was given antibiotics and within two days was better. | A doctor friend suggested she be tested for legionnaires disease because she wore the same mask every day all day long. | suggested she tested for legionnaires disease because she wore the same mask every day day legionnaires disease from the moisture bacteria |
| | **Claim span:** Turns out it WAS legionnaires disease from the moisture and bacteria in her mask. | | |
| | **Post:** If you are experiencing pregnancy complications and your doctor tells you that abortion is necessary, find a new doctor. You and your baby deserve better. Abortion is never medically necessary to save the life of the mother. | Abortion is never medically necessary to save the life of the mother. | pregnancy your Abortion is never medically necessary to save the life of the mother. |
| | **Claim span:** Abortion is never medically necessary to save the life of the mother. | | |
| Hindi | **Post:** श्रीमती अमृता देवेंद्र फडणवीस ने 'बाढ़ पुणे शहर के वास्तुकार' श्री देवेंद्र फडणवीस को खेदजनक स्थिति दिखाई। उन्होंने पुणे में पिछले 5 वर्षों में भाजपा द्वारा किए गए कार्यों को सामने लाया। धन्यवाद अमृता। एक आम पुनेकर। *(Mrs. Amrita Devendra Fadnavis shows sorry state of affairs to 'Architect of Flooded Pune City' Mr. Devendra Fadnavis. He brought out the work done by BJP in Pune in the last 5 years. Thanks Amrita. A common Punekar.)* | अमृता देवेंद्र फडणवीस ने 'बाढ़ पुणे शहर के वास्तुकार' श्री देवेंद्र फडणवीस को खेदजनक स्थिति दिखाई। *(Amrita Devendra Fadnavis showed the sorry state of affairs to the 'architect of flood-filled Pune city' Mr. Devendra Fadnavis.)* | अमृता देवेंद्र फडणवीस ने 'बाढ़ पुणे शहर के वास्तुकार' श्री देवेंद्र फडणवीस को खेदजनक स्थिति दिखाई। उन्होंने पुणे में पिछले 5 वर्षों में भाजपा द्वारा किए गए कार्यों को सामने लाया। *(Amrita Devendra Fadnavis showed the sorry state of affairs to the 'architect of flood-filled Pune city'. He brought out the work done by BJP in the last 5 years in Pune.)* |
| | **Claim span:** श्रीमती अमृता देवेंद्र फडणवीस ने 'बाढ़ पुणे शहर के वास्तुकार' श्री देवेंद्र फडणवीस को खेदजनक स्थिति दिखाई। *(Smt. Amrita Devendra Fadnavis shows sorry state of affairs to 'Architect of Flooded Pune City' Mr. Devendra Fadnavis. )* | | |
| | **Post:** Indonesia का जहाज़ मुसाफीर के साथ बाल बाल बचा #Alhamdulillah अल्लाह का शुक्र है *(Indonesia's plane narrowly escaped with passengers. #Alhamdulillah, thanks to Allah.)* | Indonesia का जहाज़ मुसाफीर के साथ बाल बाल बचा। #Alhamdulillah अल्लाह का शुक्र है *(Indonesia's plane narrowly escaped with passengers. #Alhamdulillah Thanks to Allah.)* | Indonesia का जहाज़ मुसाफीर के साथ बाल बाल बचा #Alhamdulillah *(Indonesia's plane narrowly escaped with passengers. #Alhamdulillah)* |
| | **Claim span:** Indonesia का जहाज़ मुसाफीर के साथ बाल बाल बचा *(Indonesia's plane narrowly escaped with passengers.)* | | |

Table 6: Error analysis of the GPT-4 model and the Multilingual mDeBERTa model on English and Hindi test instances from the X-CLAIM dataset. The social media post and the gold claim span are shown in the second column. The predicted claim spans for both models are provided in the third and fourth columns, respectively. The English translations for the Hindi examples are given inside parenthesis in *italics*.

# 7 Conclusion and Future Work

We proposed a novel automated data annotation methodology for multilingual claim span identification. Using it, we created and released a new dataset called X-CLAIM, which consists of real-world claim spans, and social media posts containing them, collected from numerous social media platforms in six languages: English, Hindi, Punjabi, Tamil, Telugu, and Bengali. Using state-of-the-art multilingual models, we established strong baselines based on encoder-only and generative language models. Our experiments demonstrated the benefits of multilingual training when compared to other cross-lingual transfer methods such as zero-shot transfer, or training on the translated data, from a high-resource language like English.

We observed lower performance for GPT-style generative LLMs when compared to smaller fine-tuned encoder-only language models and we discussed their error analysis in the spirit of improving the LLMs on this task.

Our work opens many important research questions: (1) How to obtain real-world claims without relying on fact-checkers analysis? (2) How to improve the understanding of LLMs about claims and social media in low-resource languages? (3) How to automatically curate multiple check-worthy claims made in the post? (4) How to improve the evaluation metric for the mCSI task? and (5) How to expand the CSI task to other low-resource languages? We plan to address these research questions in future work.

## Limitations

Our X-CLAIM dataset for the mCSI task is limited to six languages. We do not know how well the developed systems will perform in languages that are not considered in this work. Moreover, the proposed dataset handles only the primary claim in the given social media post and ignores any other potentially check-worthy claims that the post might contain. In practice, the post may contain multiple check-worthy claims.

## Ethics

**Broader Impact:** Our models and data will help fact-checkers filter out extraneous information, thus saving them significant amounts of time and resources.

**Data:** We place the utmost importance on user privacy. As a result, we have no intention of disclosing any information about the users. The data we curated is solely for research purposes, ensuring that user confidentiality and privacy are protected.

**Environmental Impact:** It is critical to acknowledge the environmental consequences of training large language models. In our case, we mitigate this concern to some extent by focusing primarily on fine-tuning pretrained models rather than training them from scratch.

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

# *Lost in Translation, Found in Spans*:
## Identifying Claims in Multilingual Social Media
### (Appendix)

## A  Data Collection

Various fact-checking websites analyze social media posts, news articles, and other information sources that may spread misleading information. We confine our data collection to those websites that meet the following requirements. First, the website should have fact-checked numerous social media posts, at least 100, so that we can have a reasonable-sized dataset. Second, it should have investigated posts containing text. We find that many social media posts investigated by fact-checkers have their claim encapsulated in another modality, such as image or video, than text. The fact-checkers manually find the claims made in the posts, which we call as *normalized* claim. Our last requirement is that the fact-checking website should provide the normalized claim on the webpage of the fact-checked post.

We find that there are only a couple of fact-checking websites that have investigated social media posts in low-resource languages and that meet the requirements discussed above. The website names, along with the number of fact-checked posts scraped from them, are reported in Table 7. For English, we collect data from ThipMedia,[9] FullFact,[10] Snopes,[11] PolitiFact,[12] Factly,[13] and Vishvasnews.[14] We use Vishvasnews for the remaining languages along with Aajtak[15] for Hindi alone. We find that there are relatively fewer posts in Telugu and Bengali than in other languages, highlighting the difficulty in creating data for these extremely low-resource languages.

We recognize the structure of the webpage for each fact-checking website and write rules (e.g., regular expressions) to collect the post text and the normalized claim. Once the post text and the normalized claim are collected, we pass the pair through various noise removal filters so that the noisy instances (like the ones that do not meet our requirements but dodged the previous steps) are removed from the data. These include removing

| Language | # Posts |
|---|---|
| English | 17,337 |
| Hindi | 2,378 |
| Punjabi | 1,262 |
| Tamil | 319 |
| Telugu | 261 |
| Bengali | 167 |

Table 7: Number of fact-checked social media posts collected from numerous fact-checking initiatives in six languages with the help of web-scraping API.

when the post text or the claim contains words like *video*, फोटो, *reel*, etc. We find that this rule is almost always correct. Further, we remove the data points when the length of the post or claim is less than 3 words, omitting the erroneously scraped text, or more than 700 words, more like news articles. These filtering steps remove only 2.5% of the total data collected, averaged across six languages.

## B  Model Training Details

We train our models using the Adam optimizer (Kingma and Ba, 2015) with weight decay of 0, $\beta_1 = 0.9$ and $\beta_2 = 0.999$. All experiments are carried out on a single A100 (40 GB) GPU. We use and adapt the code of Mittal and Nakov (2022) for our task. The models are trained with three different random seeds and we report the median of three evaluation runs since we observed a high variation of scores across the runs.

We do hyperparameter tuning for the learning rate and the batch size over the English data and use the same hyperparameters over the data of the remaining five languages. Driven by the motivation that the base transformer model is pretrained on a large corpus of text and requires less training, we use a smaller learning rate of 1e-5 for it, but a slightly bigger learning rate of 3e-4 for the token-classifier network. We use a batch size of 32 for training mBERT and mDeBERTa whereas a smaller batch size of 16 for the larger model, XLM-R. The maximum sequence length for the three encoder-only language models is set to 512 to avoid initializing and training new positional embeddings. We use early stopping with a patience of 7 epochs to find the best model checkpoint as per the best F1 score over the development set.

---

[9] https://www.thip.media/
[10] https://fullfact.org/
[11] https://www.snopes.com/
[12] https://www.politifact.com/
[13] https://factly.in/
[14] https://www.vishvasnews.com/
[15] https://www.aajtak.in/

The development set is set differently in various training methodologies. For MONOLINGUAL models, we use the development data in the target language, whereas, for MULTILINGUAL models, we combine and utilize the development sets of all languages. The *translate-train* models use the development data in the target language when available (Hindi, Punjabi, and Tamil) and use the translated English development set for Telugu and Bengali.

We provide the number of trainable parameters of the pretrained encoder-only language models in Table 8. For training on English data, XLM-R consumes nearly 1 hour of GPU runtime whereas mBERT and mDeBERTa take nearly 0.5 hours.

| Model | # Trainable Parameters |
|---|---|
| mBERT | 177,854,978 |
| mDeBERTa | 278,220,290 |
| XLM-R | 559,892,482 |

Table 8: Number of trainable parameters in the pretrained encoder-type language models.

## C Modelling Details

The encoder-only language models are trained in the frame of sequence tagging task where the model needs to predict the correct label for each token in the post text. A randomly initialized feedforward neural network is placed on top of the pretrained encoder as a token-classifier network. It takes as input the contextualized token embeddings (output by the pretrained encoder) and results in the probability distribution over the label space. The cardinality of the label space depends on how the tokens are encoded.

| Encoding | Precision | Recall | F1 |
|---|---|---|---|
| IO | 70.79 | **84.00** | **73.61** |
| BIO | 72.22 | 82.00 | 73.52 |
| BEO | 69.23 | 68.83 | 61.56 |
| BEIO | **72.28** | 80.38 | 72.30 |

Table 9: Performance comparison (in %) of encoding schemes on the English test set in X-CLAIM dataset.

We experiment with token-level encoding schemes: *IO*, *BIO*, *BEO* and *BEIO*. We train four XLM-R models, one with each encoding, on the English training data in X-CLAIM dataset and compare their performance on the English test set in X-CLAIM. The scores are reported in Table 9: *IO* encoding shows the best F1 performance among different encoding schemes.

## D Prompting the Large Language Models (LLMs) in GPT series

We use OpenAI API and evaluate text-davinci-003 (T-DV3), gpt-3.5-turbo (GPT-3.5) and gpt-4-0314 (GPT-4) models on multilingual claim span identification task through prompting. The prompts used in zero-shot prompting are provided in Figure 5. The decoding temperature is set to 0 and we use the default maximum response length. All the GPT series LLMs were prompted from Oct 16, 2023 to Oct 22, 2023.

Identify the central claim in the given text.
Post: <social media post text>
Claim:

Extract the central claim in the given text.
Post: <social media post text>
Claim:

Extract the central claim span in the given text.
Post: <social media post text>
Claim:

Extract the central claim in the given text in <language>.
Post: <social media post text>
Claim:

Figure 5: Zero-shot prompting with four prompts: IDENTIFY (first row), EXTRACT (second row), SPAN (third row), and LANGUAGE (last row). For a given social media post, its text and its language (in LANGUAGE prompt) are placed inside the prompt as shown by < >.

Extract the central claim in the given text in <language>.
Post: <training example #1: social media post text>
Claim: <training example #1: claim span>

Extract the central claim in the given text in <language>.
Post: <training example #2: social media post text>
Claim: <training example #2: claim span>

Extract the central claim in the given text in <language>.
Post: <training example #3: social media post text>
Claim: <training example #3: claim span>

Extract the central claim in the given text in <language>.
Post: <social media post text>
Claim:

Figure 6: In-context learning using LANGUAGE prompt with three training examples. For each training example, the social media post's text, the claim span and its language are placed inside the prompt as shown by < >.

| Approach | English | | Hindi | | Tamil | | Telugu | |
|---|---|---|---|---|---|---|---|---|
| | P | R | P | R | P | R | P | R |
| awesome-align | 64.08 | **95.27** | 73.62 | **89.00** | 74.37 | **99.85** | 77.01 | **96.65** |
| +ROUGE-F1 | 75.39 | 78.20 | **81.61** | 84.55 | 79.46 | 94.90 | 84.25 | 80.99 |
| +METEOR | 80.27 | 80.90 | 80.91 | 83.53 | **79.91** | 94.70 | 86.48 | 83.21 |
| +BERTScore-F1 | 80.45 | 82.23 | **81.61** | 84.55 | **79.91** | 94.70 | **88.20** | 83.44 |
| +BERTScore-Recall | **84.62** | 86.87 | **81.61** | 84.55 | **79.91** | 94.70 | 87.71 | 84.33 |

Table 10: Precision (P) and recall (R) scores (in %) of the automated annotation (Section 3.2) when using different sentence similarity measures during sentence selection. We use awesome-align alone for Punjabi and Bengali: 77.57% precision and 92.92% recall in Punjabi whereas 70.78% precision and 97.68% recall in Bengali.

| Model | English | | | Hindi | | | Punjabi | | | Tamil | | | Telugu | | | Bengali | | |
|---|---|---|---|---|---|---|---|---|---|---|---|---|---|---|---|---|---|---|
| | P | R | F1 | P | R | F1 | P | R | F1 | P | R | F1 | P | R | F1 | P | R | F1 |
| GPT-4 | **75.28** | 77.80 | **73.49** | **79.42** | 91.06 | **82.49** | **73.49** | **91.20** | **77.58** | **79.12** | 81.99 | 77.92 | **86.11** | 71.31 | 75.54 | 71.88 | 81.79 | 73.17 |
| mDeBERTa | 72.25 | **80.90** | 73.01 | 75.94 | **92.66** | 80.87 | 70.62 | 90.99 | 76.27 | 78.21 | **88.69** | **80.29** | 82.34 | **87.10** | **82.92** | **77.11** | **85.89** | **79.24** |
| Δ | ↓-3.03 | ↑3.10 | ↓-0.48 | ↓-3.48 | ↑1.60 | ↓-1.62 | ↓-2.87 | ↓-0.21 | ↓-1.31 | ↓-0.91 | ↑6.70 | ↑2.37 | ↓-3.77 | ↑15.79 | ↑7.38 | ↑5.23 | ↑4.1 | ↑6.07 |

Table 11: Performance comparison (in %) of GPT-4 and MULTILINGUAL mDeBERTa model on X-CLAIM dataset. Δ denotes the difference between the performance of MULTILINGUAL mDeBERTa and GPT-4.

| Model | English | | | Hindi | | | Punjabi | | | Tamil | | | Telugu | | | Bengali | | |
|---|---|---|---|---|---|---|---|---|---|---|---|---|---|---|---|---|---|---|
| | P | R | F1 | P | R | F1 | P | R | F1 | P | R | F1 | P | R | F1 | P | R | F1 |
| CURT | **74.96** | **83.96** | **75.38** | 76.55 | 75.50 | 71.94 | **73.60** | 74.13 | 70.32 | 76.69 | 80.91 | 75.35 | 82.96 | **82.63** | 79.68 | 74.81 | 74.76 | 69.53 |
| X-CLAIM | 74.28 | 80.72 | 73.79 | 75.71 | **91.18** | **80.08** | 73.18 | **88.87** | **76.78** | **77.68** | 80.97 | **75.44** | 84.42 | 74.91 | 76.25 | **76.42** | **79.49** | **75.88** |

Table 12: Performance comparison (in %) of the two models: mDeBERTa model trained on CURT dataset (first row) and English MONOLINGUAL mDeBERTa model trained on the English data in X-CLAIM dataset (second row).