# OpenReview forum: "$\textit{Lost in Translation, Found in Spans}$: Identifying Claims in Multilingual Social Media"
_EMNLP/2023/Conference — EMNLP 2023 Main_

### Official Review · Reviewer_G3UE · 2023-07-27

**Soundness:** 3

**Excitement:**

4: Strong: This paper deepens the understanding of some phenomenon or lowers the barriers to an existing research direction.

**Paper Topic And Main Contributions:**

The authors focus on the task of claim identification, seeks to label spans of text that contain a claim. They build automatically construct a multilingual corpus of claims by matching manually written claims to the sentence they came from in the linked document.   They do so for English and a set of Indian languages.

They train several transformer-based sequence labelling models, all using one of mBERT, mDeBERTa, and XLM-R.  They also include several training approaches, including monolingual training, multilingual training, English-only training and training translated from English.  They results do not show a consistent pattern across languages, although the multilingual models tend to work better than the multilingual ones do.

The authors also explore the performance of GPT models, and find that the best performing model (GPT-3.5-turbo) still performs worse than the best performing translation baseline.  The authors then do a small qualitative analysis of the paper.

**Questions For The Authors:**

1) In table 2, I assume the bolded numbers are the best performing models.  However, in Hindi F1 and Tamil F1, there are better performing models not bolded (mono mDeBERTa and multi mDeBERTa respectively).  Also, do the results consist of multiple runs?
2) On page 7, line 461, the authors make a claim about why GPT performs worse than their transformer based models.  However, in the error analysis section, the authors seem to point to different aspects that each model gets worse or better than the other.  Therefore, I'm not sure the earlier statement is justified?  The lower GPT performance could be due to a number of factors (e.g. less in-language training data, the need for more prompt engineering) that go beyond the models inability to reason.

**Reasons To Accept:**

1) The task is important to improving access to fact claim verification on the internet; this is made more important by the fact they include a variety of non-English languages.
2) The paper is generally easy to understand (despite a few confusing sentences), and the authors propose to release the dataset.
3) They compare a wide variety of baselines, including models that they train and pretrained approaches.
4) Their annotation approach is generalizable; potentially for other data sources to be constructed in different languages.

**Reasons To Reject:**

1) While the automated annotation approach is understandable, I wonder if the authors deal with the ways this might shape the dataset.  Given that the claims that are used to find the original claim are rewritten, I wonder if this does not skew the dataset to more easily found claims.  Perhaps claims that are more subtle are not included?  I don't think any of this is disqualifying, but I'd like to see more of a discussion of it.
2) From an NLP modeling perspective (especially the span identification task), the results are in line with what previous work has shown -- multilingual training performs better than monolingual training, with zero-shot modeling often performing slightly worse (such as in the cited Wu and Dredze 2019).  However, this doesn't appear to have been shown on this specific task (looking through the related work).  But it is a finding that is of more narrow interest within the community.
3) Following with the above point, I think it would be helpful if the authors showed what concrete performance improvements can be gained by using the multilingual model over an English-trained model.  From what I can tell, the difference is usually pretty small.  Is there a concrete performance gain on a specific type of claim that justifies automatically or manually annotating data?

**Reproducibility:**

3: Could reproduce the results with some difficulty. The settings of parameters are underspecified or subjectively determined; the training/evaluation data are not widely available.

**Reviewer Confidence:**

4: Quite sure. I tried to check the important points carefully. It's unlikely, though conceivable, that I missed something that should affect my ratings.

**Typos Grammar Style And Presentation Improvements:**

I found the two sentences in the abstract beginning at line 16 hard to follow, and I'd suggest a rephrasing.

Clearly focusing on non-English languages is very important!  However, I'm not sure that many of the included languages (e.g. Hindi) can be considered low-resource...it is in relative to English certainly.  But compared to languages with much fewer speakers (e.g. Manipuri), this is a high-resource language.  I'd suggest the authors flesh out this point more in the paper.

---

> ### Author Rebuttal · Authors · 2023-08-29
>
> Thank you for your critical feedback.
>
> [Multiple claims] We agree that there may be multiple claims present in the social media post and we collected only the primary or main claim, which is authored by professional fact-checkers for veracity estimation. The remaining claims in the social media post are likely to be less important and maybe easy to find. We acknowledge this as a limitation of our work (see lines 546-549, in main manuscript). We will provide more discussion on this in our revised version.
>
> [Multilingual and monolingual training] Multilingual training has been shown to perform better than monolingual training in different NLP tasks, e.g., open knowledge base completion (Mittal et al., ACL 2023) and semantic parsing (Li et al., EACL 2021). Since claim span identification has not been studied before in multilingual setting (see lines 90-92, in main manuscript), the related works on claim span identification have only been restricted to English due to lack of datasets in other languages. Thus, this empirical finding is not shown in any of the related works, and we discuss the same in Section 5.1 of our paper.
>
> [Multilingual and English-trained model] We see large gains achieved by multilingual models over the English-trained models in Table 2. Example, mBERT-multilingual performs better than mBERT-on-En by a large margin of 10.28% Recall and 5.98% F1 (averaged over all languages except English). We believe that the high-quality training data in other languages of our dataset, when added with English training data in multilingual training, helps the model generalize better and enhance performance.
>
> We justify automatically annotating the data with the help of our discussion in Section 5.3. We compare two models, one trained on automatically-annotated English data (MOSM), and the other trained on manually-annotated data (CURT), in Figure 3. We find that the model trained on automatically-annotated data outperforms the other model by a huge margin of 7% F1 (averaged over six languages).
>
> [Question-1] Thank you for pointing out the mistake. We will correct it in the revised draft. The results consist of a single run with a fixed random seed for reproducibility (see lines 916-919).
>
> [Question-2] Thank you for highlighting the claim made in line 436. While in the error analysis section, we discussed the kind of errors made by the two models, in line 436, we wanted to mention the possible reasons. We will rephrase the claim into possible reasons and factors why the GPT model may be performing worse, including the ones you mentioned.
>
> Thank you for the suggestions to improve our paper writing. We will revise the abstract for easier understanding and update the paper to focus on non-English languages.

---

### Official Review · Reviewer_bWC1 · 2023-08-02

**Soundness:** 4

**Excitement:**

3: Ambivalent: It has merits (e.g., it reports state-of-the-art results, the idea is nice), but there are key weaknesses (e.g., it describes incremental work), and it can significantly benefit from another round of revision. However, I won't object to accepting it if my co-reviewers champion it.

**Paper Topic And Main Contributions:**

This paper investigates the claim span identification task in six languages (Tamil, Hindi, Punjabi, Telugu, Bengali, and English). The authors propose a pipeline to automatically collect and annotated data (claim spans) in different languages. The authors then experiment with the dataset training claim span identification systems in monolingual and multilingual setups. The authors experiment also with cross-lingual transfer from English.

**Questions For The Authors:**

Question A: If I understand correctly, the metric (BERTScore) was chosen after a small experiment (manually annotating 45 claims, and checking all metrics' performance on this set). Since this experiment was done for English wouldn't it be possible, that BERTScore's performance for other languages is actually quite poor?

**Reasons To Accept:**

- the proposed dataset includes data in six different languages
- the experiments with monolingual and multilingual models are compelling and seem to be well executed
- the authors include an error analysis of the performance of their multilingual model

**Reasons To Reject:**

- the collected dataset is relatively small (excluding English) with the total number of datapoints ranging from 81 to 1457. Since the posts/claims are not long it would be very feasible to manually verify all claims to ensure the data is of a good quality (compensating for the small size). I might have missed it, but I think at least the test sets should be manually verified.

**Reproducibility:**

4: Could mostly reproduce the results, but there may be some variation because of sample variance or minor variations in their interpretation of the protocol or method.

**Reviewer Confidence:**

2: Willing to defend my evaluation, but it is fairly likely that I missed some details, didn't understand some central points, or can't be sure about the novelty of the work.

---

> ### Author Rebuttal · Authors · 2023-08-29
>
> Thank you for your thoughtful comments.
>
> [Dataset size] Our dataset is relatively small in languages except English due to the limited availability of fact-checked social media posts in other languages. We provide the statistics of available fact-checked posts in Table 6 (Appendix) and collect all these posts while creating our dataset. Our work is the first attempt towards claim span identification in these truly low-resource languages (due to lack of annotated data), and we believe that it will spur interest in the community to address these research challenges and build even larger datasets.
>
> [Manual verification of dataset] Based on your feedback, we ask linguistic experts in the six languages to manually annotate the claim spans in the test sets with the help of claims authored by fact-checkers. We re-evaluated our models on these manually-annotated test sets and found that the empirical findings hold, as shown in the paper. Since our annotation budget only allowed for the size of test sets, we evaluated the quality of our automatically collected train (and validation) sets. We provide the experts with 100 randomly sampled post texts (from train+val set) in each language and ask them to manually annotate the claim spans with the help of claims authored by fact-checkers. Then, we compare our automatically-annotated claim spans with their manually-annotated ones, and find a high data quality of 81.3% F1 in Hindi, 81.04% F1 in Punjabi and 89.73% F1 in Tamil. Note that for Telugu and Bengali, we only have the manually annotated test sets.
>
> [Question] We empirically answer this question. Above, we discussed the evaluation of the quality of data created using BERTScore. We use the same 100 randomly sampled post texts (from train+val set) from above and create the automatically-annotated claim spans with the other sentence-selection metrics, just like in English (Table 3). Then, we evaluate these metrics for the other languages using the 100 manually-annotated claims and find that BERTScore performs the best and shows a high average performance of 83.99% F1 over all the languages.
>
> We will revise our paper in the final version, if accepted, with the answers provided, the performance of models on the manually-annotated test sets, and the performance of all the sentence-selection metrics for the other languages.

---

### Official Review · Reviewer_oZSC · 2023-08-05

**Soundness:** 4

**Excitement:**

4: Strong: This paper deepens the understanding of some phenomenon or lowers the barriers to an existing research direction.

**Paper Topic And Main Contributions:**

This paper proposes a claim span identification dataset for English and lower resource languages (Hindi, Tamil, etc.).
The dataset is automatically created using fact check websites and the dataset quality is assessed for English.
Experiments are conducted in different settings: monolingual, multilingual training, and zero-shot cross-lingual transfer from English.
The authors also report performance of LLMs.
Results suggest joint multilingual training is effective.

**Questions For The Authors:**

- How did the authors collect post texts? (data source and methodology)

**Reasons To Accept:**

- The proposed methodology of creating dataset is interesting and seems effective
- The multilingual corpus of claim span identification is variable for researchers of fact-checking
- The authors give thorough experiments and analyses

**Reasons To Reject:**

- The quality assessment of the annotation is limited to English

**Reproducibility:**

4: Could mostly reproduce the results, but there may be some variation because of sample variance or minor variations in their interpretation of the protocol or method.

**Reviewer Confidence:**

3: Pretty sure, but there's a chance I missed something. Although I have a good feel for this area in general, I did not carefully check the paper's details, e.g., the math, experimental design, or novelty.

---

> ### Author Rebuttal · Authors · 2023-08-29
>
> Thank you for appreciating the contributions of our paper.
>
> [Dataset quality assessment] Based on your feedback, we evaluated the quality of training (and validation) data in Hindi, Punjabi, and Tamil with the help of linguistic experts and fixed guidelines. We provide the experts with 100 randomly sampled post texts (from train+val set) in each language and ask them to manually annotate the claim spans with the help of claims authored by fact-checkers. Then, we compare our automatically-annotated claim spans with their manually-annotated ones, and find a high data quality of 81.3% F1 in Hindi, 81.04% F1 in Punjabi, and 89.73% F1 in Tamil. For the remaining two languages, Telugu and Bengali, we only have test sets, and based on the feedback from R2, we manually annotate the test sets in all six languages to ensure the least noise in them. Our annotation budget only allowed for the size of test sets, and we provided quality assessment numbers of our automatically-annotated train (and validation) sets above.
>
> For manual annotation of the test sets, we ask the linguistic experts in the six languages to manually annotate the claim spans in the test sets with the help of claims authored by fact-checkers. We re-evaluated our models on these manually-annotated test sets and found that the empirical findings hold, as shown in the paper.
>
> [Question] The post text is provided in fact-checking websites, and we write regular expressions based on the structure of each website to scrape the post text (lines 228-232, in the main manuscript). Example of a regex: \((post said:?))\s*(,|:|-)?\s*((\'|")|(“|‘)))(.*(\s.*)?)(\.(\5|”|’))\. It is worth noting that the regex for each fact-checking website (Snopes, Politifact, etc) is different.
>
> If accepted, we will utilize the 9th page and add the clarifications and the data quality assessment numbers in the revised version.

---

### Meta-Review · Area_Chair_6ZvJ · 2023-09-16

**Recommendation:** 5

**Metareview:**

The paper introduces a multilingual resource (6 languages including English) for claim span identification, created by their proposed methodology. The reviewers appreciate the importance of the task, the dataset creation methodology, and the experiments conducted. Most of their concerns are addressed by the authors' responses.

---

### Decision · Program_Chairs · 2023-10-07

**Decision:**

Accept-Main

**Comment:**

The paper introduces a multilingual resource (6 languages including English) for claim span identification, created by their proposed methodology. The reviewers appreciate the importance of the task, the dataset creation methodology, and the experiments conducted. Most of their concerns are addressed by the authors' responses.